

# Two different approaches to the affective profiles model: median splits (variable-oriented) and cluster analysis (person-oriented)

Danilo Garcia[1,2,3,4], Shane MacDonald[3,5,6] and Trevor Archer[2,3]

[1] Blekinge Center of Competence, Blekinge County Council, Karlskrona, Sweden
[2] Department of Psychology, University of Gothenburg, Gothenburg, Sweden
[3] Network for Empowerment and Well-Being, University of Gothenburg, Gothenburg, Sweden
[4] Centre for Ethics, Law and Mental Health (CELAM), University of Gothenburg, Gothenburg, Sweden
[5] Center for Health and Medical Psychology (CHAMP), Psychological Institution, Örebro University, Örebro, Sweden
[6] Psychological Links of Unique Strengths (PLUS), Psychological Institution, Stockholm University, Stockholm, Sweden

Corresponding author
Danilo Garcia,
danilo.garcia@icloud.com

## ABSTRACT

**Background.** The notion of the affective system as being composed of two dimensions led Archer and colleagues to the development of the affective profiles model. The model consists of four different profiles based on combinations of individuals' experience of high/low positive and negative affect: self-fulfilling, low affective, high affective, and self-destructive. During the past 10 years, an increasing number of studies have used this person-centered model as the backdrop for the investigation of between and within individual differences in ill-being and well-being. The most common approach to this profiling is by dividing individuals' scores of self-reported affect using the median of the population as reference for high/low splits. However, scores just-above and just-below the median might become high and low by arbitrariness, not by reality. Thus, it is plausible to criticize the validity of this variable-oriented approach. Our aim was to compare the median splits approach with a person-oriented approach, namely, cluster analysis.

**Method.** The participants ($N = 2,225$) were recruited through Amazons' Mechanical Turk and asked to self-report affect using the Positive Affect Negative Affect Schedule. We compared the profiles' *homogeneity* and *Silhouette coefficients* to discern differences in homogeneity and heterogeneity between approaches. We also conducted exact cell-wise analyses matching the profiles from both approaches and matching profiles and gender to investigate profiling agreement with respect to affectivity levels and affectivity and gender. All analyses were conducted using the ROPstat software.

**Results.** The cluster approach (weighted average of cluster *homogeneity coefficients* = 0.62, *Silhouette coefficients* = 0.68) generated profiles with greater homogeneity and more distinctive from each other compared to the median splits approach (weighted average of cluster *homogeneity coefficients* = 0.75, *Silhouette coefficients* = 0.59). Most of the participants ($n = 1,736$, 78.0%) were allocated to the same profile (*Rand Index* = .83), however, 489 (21.98%) were allocated to different

profiles depending on the approach. Both approaches allocated females and males similarly in three of the four profiles. Only the cluster analysis approach classified men significantly more often than chance to a self-fulfilling profile (type) and females less often than chance to this very same profile (antitype).

**Conclusions.** Although the question whether one approach is more appropriate than the other is still without answer, the cluster method allocated individuals to profiles that are more in accordance with the conceptual basis of the model and also to expected gender differences. More importantly, regardless of the approach, our findings suggest that the model mirrors a complex and dynamic adaptive system.

Several health characteristics are associated with individuals' affectivity (*Watson & Tellegen, 1985*); consequently, both positive affect and negative affect possess some degree of explanatory value (e.g., *Clark & Watson, 1988*). In this context, Wilson and colleagues (*1998*) indicated that there is no significant correlation between positive affect and negative affect as measured by one of the most common instruments used to self-report affect, the Positive Affect Negative Affect Schedule (*Watson, Clark & Tellegen, 1988*). Moreover, each one of these dimensions (i.e., positive affect and negative affect) correlates to different personality and health attributes (*Garcia, 2011*; *Norlander, Bood & Archer, 2002*). Individuals characterized by high levels of positive affect exhibit a greater appreciation of life, more security, self-esteem, and self-confidence (*Archer, Adolfsson & Karlsson, 2008*; *Costa & McCrae, 1980*). They enjoy more social relations and assertiveness and are generally described as passionate, happy, energetic, and alert (*Watson & Clark, 1994*; *Watson & Pennebaker, 1989*). In contrast, individuals characterized by high levels of negative affect experience greater stress, strain, anxiety, and uncertainty over a wide range of circumstances and events (*Spector & O'Connell, 1994*; *Watson, Pennebaker & Folger, 1986*). In other words, these two dimensions that compose the affective system are uncorrelated from each other. However, even in the case of null correlations there might still be a nonlinear dependency between these two affectivity dimensions. For instance, from a person-centered framework these two affectivity dimensions within the individual can be seen as interwoven components with whole-system properties (*Bergman & Wångby, 2014*). The outlook of the individual as a whole-system unit is then best studied by analyzing patterns of information (*Bergman & Wångby, 2014*). Although at a theoretical level there is a myriad of probable patterns of combinations of peoples' levels of positive and negative affect, if viewed at a global level, there should be a small number of more frequently observed patterns or "common types" (*Bergman & Wångby, 2014*; *Bergman & Magnusson, 1997*; see also *Cloninger, Svrakic & Svrakic, 1997*, who explain nonlinear dynamics in complex adaptive systems).

In this line of thinking, Archer and colleagues (e.g., *Archer et al., 2007*; *Garcia, 2011*; *Norlander, Bood & Archer, 2002*; *Norlander, Von Schedvin & Archer, 2005*) coined the notion of the affective profiles by proposing four possible combinations using individuals' experience of high/low positive/negative affect: (1) high positive affect and low negative affect (i.e., the self-fulfilling profile), (2) low positive affect and low negative affect (i.e., the low affective profile), (3) high positive affect and high negative affect (i.e., the high affective profile), and (4) low positive affect and high negative affect (i.e., the self-destructive profile). During the last 10 years, research using the affective profiles model has distinguished individual differences in positive (i.e., well-being) and negative (i.e., ill-being) psychological and somatic health (e.g., *Garcia et al., 2010*; *Garcia, 2012*; *Garcia & Siddiqui, 2009a*; *Garcia & Siddiqui, 2009b*; *Garcia & Moradi, 2013*; *Garcia & Archer, 2012*; *Nima et al., 2013*; *Jimmefors et al., 2014*). Particularly, individuals with a self-destructive profile, compared to individuals with a self-fulfilling profile, experience lower subjective and psychological well-being, along with lower levels of energy, dispositional optimism, and higher levels of somatic stress, pessimism, non-constructive perfectionism, depression and anxiety, maladaptive coping, stress at the work-place, external locus of control, and impulsiveness (see among others *Archer et al., 2007*; *Bood, Archer & Norlander, 2004*; *Garcia, 2012*; *Garcia, Nima & Kjell, 2014*; *Karlsson & Archer, 2007*; *Palomo et al., 2007*; *Palomo et al., 2008*; *Schütz, Archer & Garcia, 2013*; *Schütz, Garcia & Archer, 2014*; *Schütz et al., 2013*). The most important differences, however, are discerned when individuals that are similar in one affect dimension but differ in the other dimension are compared to each other (*Garcia, 2011*). Individuals with a low affective profile (low positive affect, low negative affect), for example, report to be more satisfied with their life compared to individuals with a self-destructive profile (low positive affect, high negative affect). Hence, suggesting that high levels of life satisfaction are associated to decreases in negative affect when positive affect is low. In essence, the affective profiles model offers a nuanced representation of the composition of the affectivity system—a diametrically different representation than the notion of treating these two dimensions simply as two separate variables or summarizing them to create one mean value (*Garcia, 2011*; *Garcia, 2012*). See Fig. 1 for a compilation of findings from the last 10 years of research conducted by Archer, Garcia, and colleagues showing individual differences and similarities using the affective profiles model.

The most common approach to the categorization of individuals in four different affective profiles is by means of median splits. Basically, individuals' self-reported scores on positive and negative affect are divided into high and low in reference to the median (*Norlander, Bood & Archer, 2002*). The individuals high and low scores are then combined into the four profiles. However, since median splits distort the meaning of high and low, it is plausible to criticize the validity of this approach to create the affective profiles—scores just-above and just-below the median become high and low by arbitrariness, not by reality (*Schütz, Archer & Garcia, 2013*). That is, the median splits method is variable-oriented because it categorizes individuals in different affective profiles based on the variable's cut-off scores. A variable-oriented approach is, for instance, characterized for its focus

**High Positive Affect**

### Self-Fulfilling

- High levels of psychological well-being.

- High levels of subjective well-being: life satisfaction, high positive affect, low negative affect, and harmony.

- Low levels of ill-being: low depressive and stress symptoms and sleeping and psychophysiological problems.

- Personality: low in Neuroticism, high in Extraversion, low in Harm Avoidance, high in Persistence, high in Self-directedness, high in Cooperativeness, low in Dark Triad traits (i.e., Machiavellianism, Psychopathy, and Narcissism).

- Other important characteristics: frequently physical active, high on spiritual behavior, high in energy and locomotion ('just do it' mentality), low in assessment (rumination).

### High Affective

- High levels of psychological well-being: environmental mastery, self-acceptance, personal growth, and purpose in life.

- Low levels of psychological well-being: autonomy.

- High levels of subjective well-being: life satisfaction, high positive affect, and harmony.

- Low levels of subjective well-being: high negative affect.

- Low levels of ill-being: low depressive symptoms.

- High levels of ill-being: frequent sleeping and psychophysiological problems and high stress.

- Personality: high in Neuroticism, high in Extraversion, high in Harm Avoidance, high in Reward Dependence, high in Self-directedness, low in Self-transcendence, high in Dark Triad traits (i.e., Machiavellianism, Psychopathy, and Narcissism).

- Other important characteristics: frequently physical active, high in energy and locomotion ('just do it' mentality), high in assessment (rumination).

### Low Affective

- High levels of psychological well-being.

- High levels of subjective well-being: life satisfaction, low negative affect, and harmony.

- Low levels of subjective well-being: low positive affect.

- Low levels of ill-being: low depressive and stress symptoms.

- High levels of ill-being: high psychophysiological and sleeping problems.

- Personality: low in Extraversion, high in Emotional Stability, low in Persistence, low in Self-directedness, low in Cooperativeness, low in Dark Triad traits (i.e., Machiavellianism, Psychopathy, and Narcissism).

- Other important characteristics: not physical active, low in energy and locomotion ('just do it' mentality), high in assessment (rumination).

### Self-Destructive

- Low levels of psychological well-being.

- Low levels of subjective well-being.

- High levels of ill-being: high in depressive and stress symptoms and psychophysiological and sleeping problems.

- Personality: high in Introversion, high in Neuroticism, low in Persistence, high in Harm Avoidance, low in Self-directedness, low in Cooperativeness, high in Dark Triad traits (i.e., Machiavellianism, Psychopathy, and Narcissism).

- Other important characteristics: not physical active, low energy and locomotion ('just do it' mentality), high in assessment (rumination), low in spiritual behavior.

**Low negative affect**

**High negative affect**

**Low positive affect**

**Figure 1** Summary of the main findings during the past 10 years using the affective profiles model by Archer, Garcia, and colleagues.

on differences between individuals without considering the existence of sub-populations (*Lundh, 2015*). In this regard is plausible to suggest that because the affective profiles model is, at least in theory, person-centered, it should be operationalized using an approach that focuses on internal patterns, rather than individual differences (cf. *Lundh, 2015*).

Recently, *MacDonald & Kormi-Nouri (2013)* used person-oriented research approaches to cluster individuals depending on their self-reported affectivity and found that the four profiles emerged as originally modeled by Archer and as operationalized using the median splits approach. However, although apparently similar, we argue that these two approaches are still different in their research focus with respect to two contrasts: (a) variable versus pattern focused and (b) individual versus population focused (cf. *Lundh, 2015*). The median splits approach focuses on variables and their cut-off values in populations, thus it is a top-down procedure. A bottom-up procedure, in contrast, is the hierarchical cluster analysis, which starts by sequentially joining the most similar participants on variables of interest (e.g., positive affect and negative affect) to form groups (i.e., pattern and individual focused). A follow up relocation procedure may then use K-means cluster analysis to ensure people are assigned to a profile most similar to theirs (see *MacDonald & Kormi-Nouri, 2013*; *Kormi-Nouri et al., 2015*). In this respect, cluster analytic methods are data-driven and create profiles that are relative to each other. Data-driven methods, compared to median splits, come closer to modeling the dynamic nature of within and between group variability of individual patterns of affectivity, while the median splits procedure is static in nature—equally sized groups are pre-determined because each one of the two variables is divided in high and low using the median.

We argue further that, depending on how profiles are made (i.e., median splits vs. cluster) the model has the potential to discern differences not found before. On average, for example, women recall experiencing negative affect to a larger extent compared to men, while on average men recall experiencing positive affect to a larger extent compared to women (e.g., *Crawford & Henry, 2004*; see also *Schütz, 2015*). Despite this fact suggesting clear general differences in affectivity between men and women, past research using the median splits has not found interaction effects between the type of profile and the person's gender on well-being and ill-being (see *Garcia, 2011*). While it is plausible to suggest that the differences in affectivity between profiles overrule possible gender differences (*Garcia & Siddiqui, 2009a*; *Garcia, 2011*), it might be so that this lack of findings depends on the choice of method to create the profiles. Indeed, in contrast to the variable-oriented method (i.e., median splits), the person-oriented method (i.e., cluster analysis) has as a primary criterion that a sample is analyzed assuming it is drawn from more than one population (*Von Eye & Bogat, 2006*), for example, males and females.

In sum, the aim of this paper is to compare the most often used variable-oriented median splits approach with the person-oriented cluster analysis approach when categorizing individuals into any of the four affective profiles of the model. As a first step, we compared the homogeneity within the profiles created with the two different approaches and also whether the profiles created with each approach were distinct from each (i.e., heterogeneity between profiles). This was important because, according to the

model, people allocated to a specific profile are expected to be similar to each other and distinct to those allocated to any of the other profiles. As a second step, we compared the two procedures to see how they agreed upon classifying people with respect to their affectivity levels. As a third and final step, we compared how males and females were allocated depending on the approach used to create the profiles.

## METHOD

### Ethical statement

After consulting with the Network for Empowerment and Well-Being's Review Board we arrived at the conclusion that the design of the present study (e.g., all participants' data were anonymous and will not be used for commercial or other non-scientific purposes) required only informed consent from the participants.

### Participants and procedure

The participants ($N = 2{,}225$, age *mean* $= 31.79$, *sd.* $= 15.58$, 1,160 males and 1,065 females) were recruited through Amazons' Mechanical Turk (MTurk; https://www.mturk.com/mturk/welcome). MTurk allows data collectors to recruit participants (workers) online for completing different tasks in exchange for wages. This method of data collection online has become more common during recent years and it is an empirically tested tool for conducting research in the social sciences (see *Buhrmester, Kwang & Gosling, 2011*). Participants were recruited by the criteria of being a US-resident and the ability to read and write fluently in English. Participants were paid a wage of .50 cents (US-dollars) for completing the task and informed that the study was confidential and voluntary. The participants were presented with a battery of self-reports comprising the affectivity measure as well as questions pertaining to age and gender.

### Instrument

#### *Positive affect negative affect schedule (Watson, Clark & Tellegen, 1988)*

Participants are instructed to rate to what extent they have experienced 20 different feelings or emotions (10 positive, such as, strong, proud, interested, and 10 negative, such as, afraid, ashamed, nervous) during the last weeks, using a 5-point Likert scale (1 = *very slightly*, 5 = *extremely*). We averaged the individual items to derive participants' scores in each scale, that is, positive affect and negative affect. *Cronbach's $\alpha$*[1] in the present study were .90 for positive affect and .88 for negative affect.

### Statistical treatment

At a general level the distribution of the positive affect scores are approximately normal (*skewness* $= -.18$, *kurtosis* $= -.30$). The negative affect scores are heavily skewed on the right (*skewness* $= 1.12$, *kurtosis* $= .98$). This comes primarily from the fact that within the value range of negative affect (1–5) the median (1.70) is very close to the minimum (1). See Fig. 2 for the distribution of positive and negative affect and Figs. 3A and 3B for the mean in both affectivity dimensions for each of the profiles created with the median splits and cluster approaches.

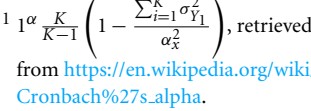

[1] $1\alpha \frac{K}{K-1}\left(1 - \frac{\sum_{i=1}^{K}\sigma_{Y_1}^2}{\alpha_x^2}\right)$, retrieved from https://en.wikipedia.org/wiki/Cronbach%27s_alpha.

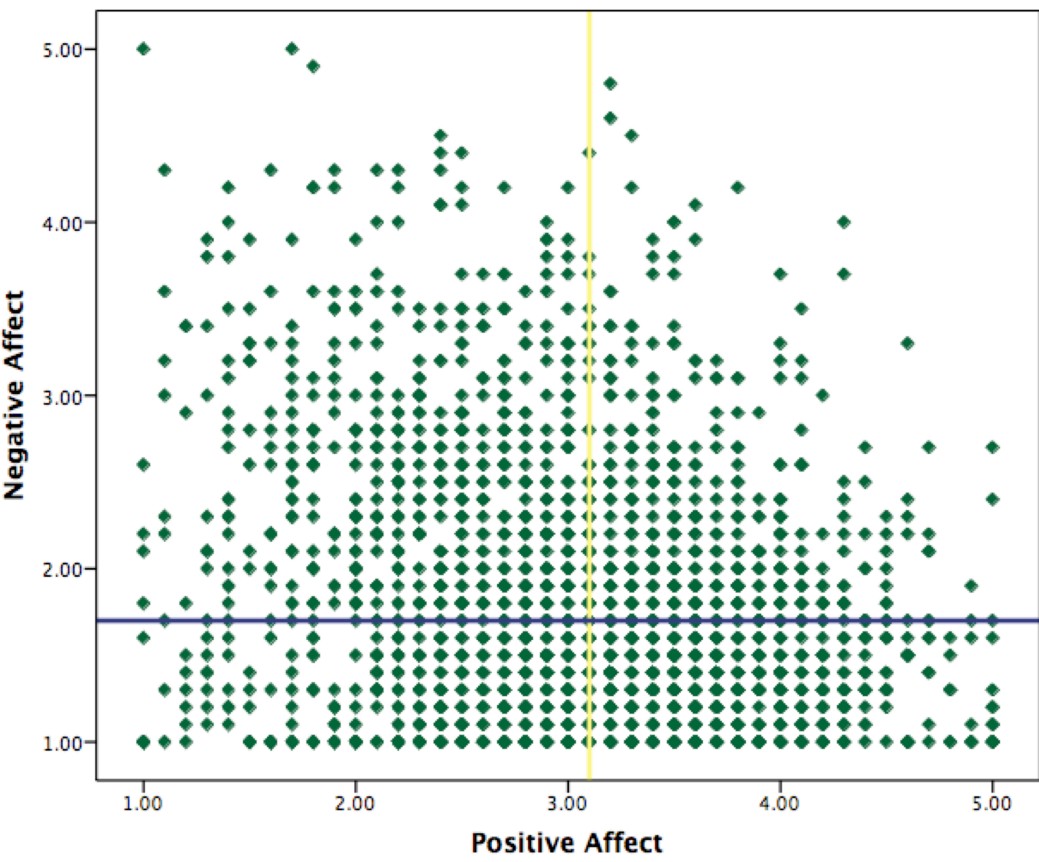

**Figure 2 Distribution of positive and negative affect.** The vertical yellow line marks the median for positive affect (3.10) and the horizontal blue line marks the median for negative affect (1.70).

### Median splits

Participants' positive affect and negative affect scores were divided into high and low as the original method used in past studies (cut-off points in the present study: low positive affect = 3.00 or less; high positive affect = 3.10 or above; low negative affect = 1.60 or less; and high negative affect = 1.70 or above). The median splits method resulted in 641 individuals with a self-fulfilling profile (351 males, 290 females), 441 individuals with a low affective profile (235 males, 206 females), 529 individuals with a high affective profile (283 males, 246 females), and 614 individuals with a self-destructive profile (291 males, 323 females). This statistical procedure was conducted in SPSS version 22.

### Cluster analysis

Ward's hierarchical cluster analysis was used to divide the sample into four groups. K-means cluster analysis used the starting points from this analysis to ensure that people ended up in a group most similar to their affective profile. The cluster analysis resulted in 781 individuals with a self-fulfilling profile (431 males, 350 females), 640 individuals with a low affective profile (336 males, 304 females), 459 individuals with a high affective profile (251 males, 208 females), and 345 individuals with a self-destructive profile (142 males,

a.

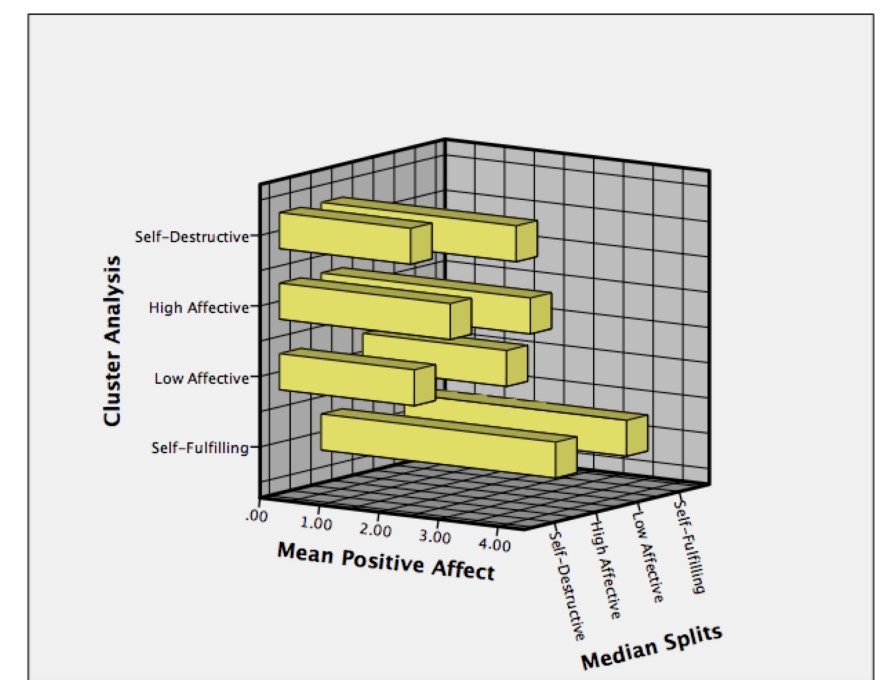

b.

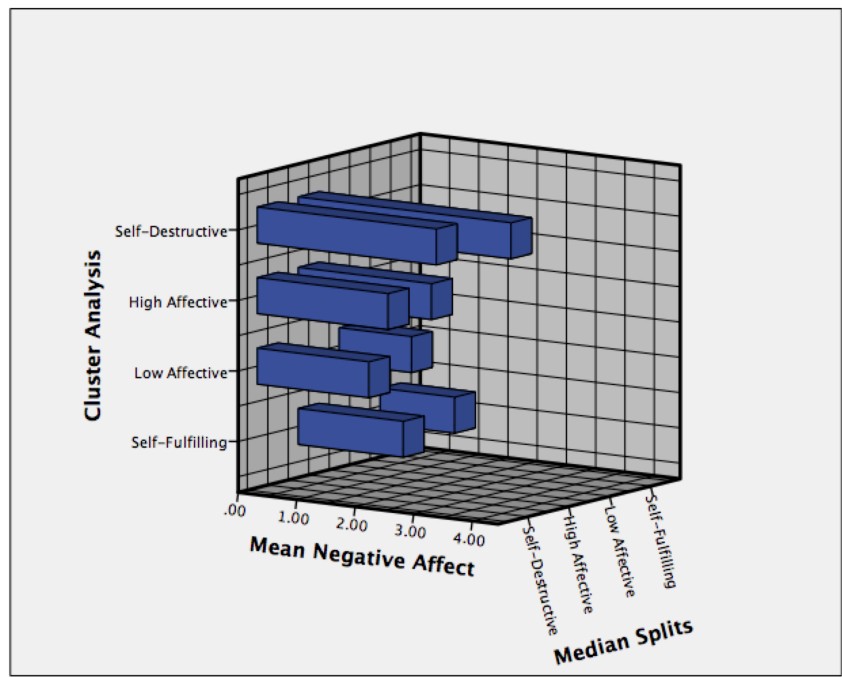

**Figure 3** Means in positive affect (A: "Joy") and negative affect (B: "Sadness") for each profile derived using the median splits and cluster analysis approaches.

**Table 1 Affective profiles pattern of standardized means for median splits and cluster approaches.**

| | Median splits | | | | Cluster | | | |
|---|---|---|---|---|---|---|---|---|
| | Prevalence (%) | Homogeneity | Positive affect | Negative affect | Prevalence (%) | Homogeneity | Positive affect | Negative affect |
| Self-fulfilling | 641 (29) | 0.41 | HIGH | low | 781 (35) | 0.46 | HIGH | (low) |
| Low affective | 441 (20) | 0.47 | low | low | 640 (29) | 0.63 | low | (low) |
| High affective | 529 (24) | 0.86 | HIGH | (HIGH) | 459 (20) | 0.53 | . | (HIGH) |
| Self-destructive | 614 (27) | 1.2 | low | HIGH | 345 (16) | 1.1 | low | HIGH$^{+++}$ |

**Notes.**

Silhouette coefficient was 0.59 for the median splits method and 0.68 for the cluster method. Weighted average of cluster homogeneity coefficient was 0.75 for the median splits method and 0.62 for the cluster method.

Simple appearance, $0.675 \leq |z| \leq 1.000$ ($p$: 16–25%).

( ), $0.44 \leq |z| \leq 0.674$ ($p$: 25–33%).

$+++$, $1.645 \leq |z| \leq 2.044$ ($p$: 2–5%).

[2] The homogeneity coefficient of a cluster is the average of the pairwise differences of cases belonging to this cluster (A Vargha, pers. comm., 2015).

[3] $s(i) = \frac{b(i)-a(i)}{\max\{a(i),b(i)\}}$, in which:

$S$, silhouette.

$i$, each single data point.

$a(i)$, the average dissimilarity of $i$ with all other data within the same cluster. That is, $a(i)$ can be interpreted as how well $i$ is assigned to its cluster (the smaller the value, the better the assignment). This allow us to define the average dissimilarity of point $i$ to a cluster c as the average of the distance from $i$ to points in c.

$b(i)$, the lowest average dissimilarity of $i$ to any other cluster, of which $i$ is not a member. The cluster with this lowest average dissimilarity is said to be the "neighboring cluster" of $i$ because it is the next best fit cluster for point $i$ (*Rousseeuw, 1987*).

203 females). This and all analyses reported under the 'Results' were conducted using the ROPstat software (*Vargha, Torma & Bergman, 2015*; http://www.ropstat.com).

# RESULTS

## Homogeneity within and heterogeneity between profiles

See Table 1 for the composition of median splits and cluster profiles. Both approaches had only one group, the self-destructive profile, that contained individuals who were dissimilar to the extent their *homogeneity coefficient*[2] value exceeded 1 (see *Bergman, Magnusson & El-Khouri, 2003*, who suggest that a *homogeneity coefficient* should ideally not exceed 1 for a homogenous grouping). On basis of the model, it is expected that individuals within each profile are similar to each other (i.e., homogeneity) and that profiles are distinctive from each other (i.e., heterogeneity). Hence, we also computed a weighted average of cluster *homogeneity coefficients* of the profiles derived using the median splits (weighted average of cluster *homogeneity coefficient* = 0.75) and cluster approaches (weighted average of cluster *homogeneity coefficient* = 0.62). In addition, we also report here the *Silhouette coefficient*,[3] which is an adequacy measure that takes into account the participants who lie within their clusters and also the ones who are merely somewhere in between clusters (*Rousseeuw, 1987*). A *Silhouette coefficient* closer to 1 might indicate that the groups are more distinct from each other (*Bergman, Magnusson & El-Khouri, 2003*). In the present sample, the cluster approach seems to generate more heterogeneous groups (*Silhouette Coefficient* = 0.68) than those profiles created using the median splits approach (*Silhouette Coefficient* = 0.59). Nevertheless, because the *Silhouette Coefficient* takes into account both the homogeneity of the clusters and the level of separation of the different clusters, the most accurate proof of heterogeneity between profiles is the differences between approaches in their weighted average of cluster *homogeneity coefficient*. One way or another, the cluster approach seems to have created profiles with greater homogeneity within the groups and also profiles that were more distinctive between each other. One important observation is that people is allocated differently depending on the approach. For example, the percentage of people being allocated in the self-destructive

**Table 2 Exact cell-wise analysis of two-way frequencies of profiles generated with the median splits and the cluster approaches.**

| | | Cluster analysis | | | |
|---|---|---|---|---|---|
| | | Self-fulfilling | Low-affective | High-affective | Self-destructive |
| **Median splits** | **Self-fulfilling** | Type | Antitype | Antitype | Antitype |
| | Observed | 641 | 0 | 0 | 0 |
| | Expected | 225.00 | 184.00 | 132.23 | 99.40 |
| | **Low-affective** | Antitype | Type | Antitype | Antitype |
| | Observed | 0 | 441 | 0 | 0 |
| | Expected | 154.80 | 126.80 | 91.00 | 68.40 |
| | **High-affective** | Antitype | Antitype | Type | Antitype |
| | Observed | 140 | 0 | 349 | 40 |
| | Expected | 185.70 | 152.20 | 109.10 | 82.00 |
| | **Self-destructive** | Antitype | Type | – | Type |
| | Observed | 0 | 199 | 110 | 305 |
| | Expected | 215.52 | 176.60 | 126.70 | 95.20 |

**Notes.**

Grey fields in diagonal highlight the cells in which there is a general agreement between approaches when allocating people to specific affective profiles. Black fields highlight the cells in which discrepancies between approaches were found. *Rand Index* = .83.

Type: the observed cell frequency is significantly greater than the expected ($p < .05$).

Antitype: the observed cell frequency is significantly smaller than the expected ($p < .05$).

– the observed cell frequency is as expected.

profile using the cluster method were 16%, while 27% were allocated in this same profile using the median splits method.

## Classification by affectivity levels between approaches

Next, we compared the two procedures to see how they agreed upon classifying people with respect to their affectivity levels using an exact cell-wise analysis. The number of people allocated in profiles formed using median splits was crossed with the number of people in profiles resulting from cluster analysis. The aim with this base model was to create a reference (i.e., an estimated expected cell frequency) to which the observed cell frequency is compared against (see *Von Eye, Bogat & Rhodes, 2006*). In short, if a specific cell contains more cases than expected under this base model, this cell indicates a relationship that exists only in this particular sector of the cross-classification, that is, it constitutes a *type*. If a cell, in contrast contains fewer cases than expected under the base model, this cell also indicates a local relationship, that is, it constitutes an *antitype* (see also *Bergman & El-Khouri, 1987*). As shown in Table 2, there is general agreement between approaches when allocating people to specific affective profiles—all cells that correspond to the same profiles indicate *types*. However, there were four sizable discrepancies between the approaches. Firstly, 199 individuals who were classified as having a self-destructive profile using the median splits procedure were allocated to a low affective profile when the cluster analysis approach was used. Secondly, 140 individuals who were allocated to a high affective profile using the median splits procedure were allocated to a self-fulfilling profile when the cluster analysis was used. The third discrepancy was that 40 individuals who were allocated to a high

**Table 3** Exact cell-wise analysis of two-way frequencies: gender and profiles generated with the median splits and cluster approach, respectively.

| Gender | Self-fulfilling | Low-affective | High-affective | Self-destructive |
|---|---|---|---|---|
| | | Median splits affective profiles | | |
| Male | – | – | – | Antitype |
| Observed (%) | 351 (54.80%) | 235 (53.30%) | 283 (53.50%) | 291 (47.40%) |
| Expected | 334.20 | 229.90 | 275.80 | 320.10 |
| Female | – | – | – | Type |
| Observed (%) | 290 (45.20%) | 206 (46.70%) | 246 (46.50%) | 323 (52.60%) |
| Expected | 306.80 | 211.10 | 253.20 | 293.90 |
| | | Cluster analysis affective profiles | | |
| Male | Type | – | – | Antitype |
| Observed (%) | 431 (55.20%) | 336 (52.50%) | 251 (54.70%) | 291 (41.20%) |
| Expected | 407.20 | 333.70 | 239.30 | 320.10 |
| Female | Antitype | – | – | Type |
| Observed (%) | 350 (44.80%) | 304 (47.50%) | 208 (45.30%) | 203 (58.80%) |
| Expected | 373.80 | 306.30 | 219.70 | 165.10 |

**Notes.**

Type (grey fields), the observed cell frequency is significantly greater than the expected ($p < .05$).
Antitype (black fields), the observed cell frequency is significantly smaller than the expected ($p < .05$).
–, the observed cell frequency is as expected.

affective profile using the median splits procedure were allocated to a self-destructive profile when the cluster analysis approach was used. The fourth and final difference was that 110 individuals who were allocated to a self-destructive profile using the median splits procedure were allocated to a high affective profile when cluster analysis was used. In sum, most of the participants ($n = 1,736$, 78.02%) were allocated to the same profile regardless of the approach being used to create the affective profiles, but 489 participants (21.98%) were allocated to different profiles depending on the approach. The *Rand Index*[4], a global measure for the overall similarity of the profiling conducted by the two approaches, was .83. The *Rand Index* computes a similarity measure between the two profiling approaches by considering all pairs of samples and counting pairs that are assigned in the same or different profiles. The *Rand Index* is ensured to have a value close to 0 for random labeling independently of the number of profiles and exactly 1 when the profiling is identical. Hence, there is a large agreement between approaches.

## Gender and the affective profiles

In a third step we examined the idea of gender having an effect on profile membership. Here, the number of males and females was crossed with the number of people in profiles resulting from each of the approaches (see Table 3). The median splits and cluster analysis approaches both allocated females to a self-destructive profile more often than chance (i.e., *type*) and males less often than chance to this very same profile (i.e., *antitype*). For the high affective and the low affective profiles, both approaches allocated males and females as expected. Nevertheless, cluster analysis differed from median splits by allocating men significantly more often than chance to a self-fulfilling profile (*type*) and females

[4] $R = \frac{a+b}{a+b+c+d} = \frac{a+b}{\binom{n}{2}}$, in which:

*a*, the number of pairs of elements in $S$ that are in the same set in $X$ and in the same set in $Y$,
*b*, the number of pairs of elements in $S$ that are in different sets in $X$ and in different sets in $Y$,
*c*, the number of pairs of elements in $S$ that are in the same set in $X$ and in different sets in $Y$, and
*d*, the number of pairs of elements in $S$ that are in different sets in $X$ and in the same set in $Y$.
Retrieved from https://en.m.wikipedia.org/wiki/Rand_index.

less often than chance to a self-fulfilling profile (*antitype*), see Table 3. Nevertheless, the proportions of males and females allocated in the different profiles seem, on visual inspection, relatively similar for both approaches (see percentages in Table 3). The greatest discrepancies between approaches in gender distributions were found in the self-destructive profile. Specifically, in the self-destructive profile created using the median splits method, the proportions within the profile were: 47.40% males and 52.60% females; while the proportions were: 41.20% males and 58.80% within the self-destructive group created using the cluster method.

## DISCUSSION

The present study set out to compare two approaches (median splits vs. cluster analysis) to making profiles as derived by the notion of the affectivity system as composed of two dimension: positive affect and negative affect. In both approaches one and the same profile showed lower homogeneity, namely, the self-destructive. There were, however, three main differences: (1) both the homogeneity within profiles and the heterogeneity between profiles were significantly larger for those profiles created with the cluster method, (2) although most of the participants ($n = 1,736$, 78.02%) were allocated to the same profile regardless of the approach and a large level of agreement between approaches, a total of 489 participants (21.98%) were allocated to different profiles, (3) and while both methods allocated males and females similarly across three of the four profiles, the methods differed in the way males and females were classified within the self-fulfilling profile. We suggest that these three differences mirror that the median splits method derives profiles focusing on variables, while the cluster method has a pattern focus that assumes the existence of data clusters, which may or may not correspond to any real subpopulations such as males and females.

According to the model (*Archer, Adolfsson & Karlsson, 2008*; *Norlander, Bood & Archer, 2002*; *Garcia, 2011*), the notion of the affectivity system as composed by two independent dimensions suggests four profiles comprising individuals who have different levels of affectivity *between* the profiles (i.e., heterogeneity), but have similar levels of affectivity *within* the profiles (i.e., homogeneity). The cluster approach generated profiles of individuals who were both more similar within (i.e., homogeneous) and more distinct from each other (i.e., heterogeneous), thus, showing that this approach is more in concordance to the theoretical basis of the affective profiles model (cf. *Keren & Schul, 2009*). However, it is plausible to question why both approaches show that individuals within the self-destructive profile are dissimilar from each other. Importantly, low levels of positive affect and high levels of negative affect do not only characterize the self-destructive profile; this affectivity combination is also a good measure of depression (*Clark & Watson, 1991*). Individuals struggling with depression have, indeed, been found to be part of a rather heterogeneous group (*Goldberg, 2011*). For example, although clustered together, depression patients may show opposite symptoms, such as, psychomotor retardation, hypersomnia and weight gaining in some cases, while agitation, bad sleep, and weight loss in another cases (*Lux & Kendler, 2010*). In other words, both approaches seem to mirror the heterogeneity, rather

than the homogeneity, within a group of individuals who experience low levels of positive affect and high levels of negative affect (i.e., the self-destructive profile). Nevertheless, this might also imply that a four-profiles solution is not the best fit for the model.

Interestingly, 309 individuals who were allocated to the self-destructive profile using the median splits method were allocated to either the low affective ($n = 199$) or the high affective profile ($n = 110$) when the cluster method was used. Moreover, 180 individuals who were allocated to the high affective profile using the median splits method were allocated to either the self-fulfilling ($n = 140$) or the self-destructive profile ($n = 40$) when the cluster method was used. All these "moving" individuals ($n = 389$) constitute 21.98% of the total population in the present study. This "movement" might suggest that individuals who are at the very end of being high or low in relation to the median in any of the affectivity dimensions *tip over* when the cluster method is used. For example, the 199 individuals who "moved" from the self-destructive profile (i.e., low positive affect/high negative affect) to the low affective profile (low positive affect/low negative affect) are individuals who certainly are low in positive affect; but that are probably closer to the median in negative affect. In contrast, the 110 individuals who "moved" from the self-destructive profile (i.e., low positive affect/high negative affect) to the high affective profile (i.e., high positive affect/high negative affect) are individuals who certainly are high in negative affect; but are probably far way from the median in positive affect. This is, for instance, in line with our finding suggesting that the self-destructive group was the less homogeneous across both approaches. Nevertheless, most of the participants ($n = 1,736$, 78.02%) were allocated to the same profile regardless of the approach being used. We suggest that this agreement in four possible affectivity combinations reflects the affective profiles model as being conceptually person-oriented. At the very least, it shows that it might be reasonable to suggest four "common types" derived of the combination of high/low positive and negative affectivity levels.

Also in this line, both methods allocated males and females similarly across three of the four profiles. Specifically, both approaches allocated females and males neither higher nor lower than expected in both the low affective and high affective profiles. In addition, both approaches allocated females to a self-destructive profile more often than chance (i.e., *type*) and males less often than chance to this very same profile (i.e., *antitype*). This specific finding across the self-destructive profiles is in accordance to differences in affectivity between males and females (for a review see *Schütz, 2015*). Consequentially, this pattern also implies that the opposite should be expected, that is, with respect to the gender distribution within the self-fulfilling profile. However, only when the cluster method was applied, were males more often than expected allocated to the self-fulfilling profile (i.e., *type*) and females were less often than expected allocated to the self-fulfilling profile (i.e., *antitype*). In other words, in contrast to the median splits method, the cluster method seems to allocate individuals in profiles that mirror gender differences found in the current literature (e.g., *Schütz, 2015*).

Nonetheless, the proportions of males and females within each profile were rather similar between approaches. Remarkably, the differences in proportions were largest

for the self-destructive profile (41.20% males and 58.80% females using the cluster method, 47.40% males and 52.60% females using the median split method) and not for the self-fulfilling profile—the only profile in which the approaches differed in the gender-pattern detailed above. Moreover, the 309 individuals who were allocated to the self-destructive profile using the median splits method, and that were allocated to either the low affective or the high affective profile when the cluster method was used, do not seem to have altered the proportions of males and females in the low affective and high affective profiles created with the cluster method. Certainly, the literature suggests that, compared to males, females have a tendency to experience high affectivity in both dimensions (*Diener et al., 1991*; *Diener, Sandvik & Pavot, 1991*; *Garcia & Erlandsson, 2011*; *Schimmack & Diener, 1997*). Still, 21.98% of the population in the present study was allocated differently depending of the approach. We suggest that, besides gender, other variables of interest in future studies might be ethnicity, religious affiliation, and motivation. After all, these shape the emotions people want to feel—that is, their "ideal affect" (*Scollon et al., 2009*; *Tsai, Knutson & Fung, 2006*; *Tsai, Miao & Seppala, 2007*; *Tsai et al., 2007*; *Cloninger & Garcia, 2015*).

## Limitations and further suggestions

Besides the limitations presented by a cross-sectional design (e.g., the inability to suggest in which direction participants "move" or are allocated from one profile to another depending on the approach), it is reasonable to discuss the data collection method used here (i.e., through MTurk). Some aspects related to this method might influence the validity of the results, such as, workers' attention levels, cross-talk between participants, and the fact that participants get remuneration for their answers (*Buhrmester, Kwang & Gosling, 2011*). Nevertheless, a large quantity of studies show that data on psychological measures collected through MTurk meets academic standards, is demographically diverse, and also that health measures show satisfactory internal as well as test-retest reliability (*Buhrmester, Kwang & Gosling, 2011*; *Horton, Rand & Zeckhauser, 2011*; *Shapiro, Chandler & Mueller, 2013*; *Paolacci, Chandler & Ipeirotis, 2010*). In addition, the amount of payment does not seem to affect data quality; remuneration is usually small, and workers report being intrinsically motivated (e.g., participate for enjoyment) (*Buhrmester, Kwang & Gosling, 2011*).

In another more important matter, the choice of approach (i.e., median splits vs. cluster) to categorize individuals in different affective profiles might depend of the distribution of the data at hand. For instance, in the present sample it seems to be evident that the median splits method does not yield naturally separable four profiles because it cuts the whole sample in cut-off points where cases are closest to each other. Due to this, cases being very close to each other may be sorted into different profiles. In addition, albeit we were interested into test the four-profile solution suggested by Archer, even the four-cluster structure created with the cluster analysis does not seem to be a natural good solution. From a theoretical point of view, future studies might strive to find the best structure of cluster analysis and compare this to the four profiles originally suggested by Archer

and colleagues. Another solution to this data-distribution problem would be to use an amalgamation of the methods. If the data have a symmetric and unimodal distribution in a dimension, it is reasonable to use median splits in that dimension. If the data has a bimodal distribution that can be well separated into two clusters in the other dimension, it is reasonable to use clustering in that dimension. In other words, the choice between median splits and clustering is probable best though as dimension-wise data dependent. Yet, another solution would be to create three categories with two cut-off points (e.g., with quartiles 1 and 3): one category in the middle and two on the tails.

Furthermore, future studies need to assess empirical differences in, for example, health measures between profiles created with the different approaches. Future studies should also compare the profiles created with different approaches using person-oriented techniques. In the present study, for example, we used exact cell-wise analyses to investigate if gender explained the allocation of individuals to different profiles. Although the same can be done using education level, ethnicity, and religious affiliation, and other variables of interest; there is an increasing amount of person-centered methods that can be used as detailed in recent literature (see among others *Bergman & Lundh, 2015*; *Valsiner, 2015*; *Lundh, 2015*; *Molenaar, 2015*; *Laursen, 2015*; *Asendorpf, 2015*; *Von Eye & Wiedermann, 2015*; *Aunola et al., 2015*; *Vargha, Torma & Bergman, 2015*; *Baker, 2015*).

### Concluding remarks

Our results suggest that the cluster method allocates individuals to profiles that are more in accordance with the conceptual basis of the model and also to expected gender differences. The question whether one approach is more appropriate than the other is still without answer, but the present study is only a first step in the development of the affective profiles model beyond the past 10 years of research. More importantly, regardless of the approach, the model of the affective system proposed by Archer and colleagues at the beginning of this century, actually mirrors a complex adaptive system. In other words, it is an affective system that is dynamic both between and within individuals and presents a probabilistic and exponentially complex reality.

> *"Flowers are restful to look at. They have neither emotions nor conflicts."*
> —*Sigmund Freud*

## ACKNOWLEDGEMENTS

First of all we would like to express our gratitude to both reviewers, Professor Andras Vargha and Professor Jingyi Jessica Li, for their comments and suggestions, which helped us to greatly improve the original manuscript. We would like to thank Sophia Isabella Garcia Rosenberg and Linnéa Mercedes Garcia Rosenberg for the inspiration to Figs. 3A ("Joy") and 3B ("Sadness"). Professor Andras Vargha's suggestion for Fig. 2 is also most appreciated as well as his help providing the ROPstat software.

### Funding

This study was supported by AFA Insurance (Dnr 130345). The funders had no role in study design, data collection and analysis, decision to publish, or preparation of the manuscript.

### Grant Disclosures

The following grant information was disclosed by the authors:
AFA Insurance: Dnr 130345.

### Competing Interests

The authors declare there are no competing interests. Danilo Garcia is the Director of the Blekinge Centre of Competence, which focuses on education, research, and development of public health and healthcare.

### Author Contributions

- Danilo Garcia conceived and designed the experiments, performed the experiments, analyzed the data, contributed reagents/materials/analysis tools, wrote the paper, prepared figures and/or tables, reviewed drafts of the paper.
- Shane MacDonald analyzed the data, contributed reagents/materials/analysis tools, wrote the paper, prepared figures and/or tables, reviewed drafts of the paper.
- Trevor Archer reviewed drafts of the paper.

### Human Ethics

The following information was supplied relating to ethical approvals (i.e., approving body and any reference numbers):

After consulting with the Network for Empowerment and Well-Being's Review Board we arrived at the conclusion that the design of the present study (e.g., all participants' data were anonymous and will not be used for commercial or other non-scientific purposes) required only informed consent from the participants.

### Data Availability

The raw data is available upon request to the Network for Empowerment and Well-Being, lead researcher Danilo Garcia: http://ltblekinge.se/Forskning-och-utveckling/ Blekinge-kompetenscentrum/Summary-in-English/.

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
