# Peer review of "Two different approaches to the affective profiles model: median splits (variable-oriented) and cluster analysis (person-oriented)"

_PeerJ, doi:10.7717/peerj.1380_

## Round 0.1 · original submission · Major Revisions

· Academic Editor

Major Revisions

The reviewers have raised several substantial concerns regarding the manuscript in its current form. These include justification for the statistical approaches used and that the framing of the manuscript could be clearer to appeal to a greater audience.

·

Basic reporting

It is OK.

Experimental design

The aim of the paper was to compare the median split approach with a person-oriented cluster analysis approach in an empirical study of Positive Affect Negative Affect Schedule. The applied dependent variables were the scales of Positive Affect (PA) and Negative Affect (NA).
The paper is interesting and worth to be published after some improvements. My major concern is as follows. Both the median split and the cluster analysis are based on the bivariate distribution of variables PA and NA. Looking at the scatter plot (prepared by means of the downloaded SPSS data file) of PA and NA it seems to be evident that the median split method does not yield naturally separable four categories in this case because it cuts the whole sample in cut-off points where cases are the most dense, that is the closest to each other. Due to this, cases being very close to each other often may be sorted into different categories. For this reason the paper should comment this inadequacy of the median split method.
Also in the cluster analysis a four-cluster structure does not seem to be a natural good solution, albeit I understand that the reader can be interested in the comparison of two four-category profiles. Theoretically, however, I would suggest finding the best structure of cluster analysis and comparing this to the four-fold categorization of the median split method.

Validity of the findings

It is generally OK, special comments are as follows.

P. 11/ line 196: The authors could mention here that the Positive affect (PA) and Negative affect (NA) scales had been derived by averaging the individual items. This may clarify why the medians were so low (3-3.1 and 1.8-1.9; see page 12, lines 200-201). It would be fine to have some descriptive statistics of PA and NA. By means of the downloaded data file I got for the medians 3.1 and 1.7 and had the impression that PA is approximately normal (skewness = -.18, kurtosis = -.30), but NA is obviously not normal (skewness = 1.12, kurtosis = .98), being heavily skewed on the right. This comes primarily from the fact that within the value range of NA (1-5) the median (1.7) is very close to the minimum (1).

P. 12/line 198: Regarding that the median for NA is 1.7, it is not clear for me why 1.8 and 1.9 were applied for NA in the median split method.

P. 13/lines 221-222 (similar problem with P. 16/lines 291-292): The expression “more heterogeneous” is incorrect. On the one hand, cluster structures with larger Silhouette coefficient values are generally more homogeneous. On the other hand, the general heterogeneity of a cluster structure can be better assessed with a weighted average of the homogeneity coefficients of the individual clusters, because the Silhouette coefficient is an adequacy measure that takes into account both the homogeneity of the clusters and the level of separation of the different clusters (Rousseeuw, 1987). And if the authors would like to compare the overall separation level of the profiles created by the median split approach and by the cluster analysis, there are some other well-known adequacy measures, like the explained variance % (EESS%, see Vargha, Torma & Bergman 2015), or a weighted average of the homogeneity coefficients of the individual clusters or median split categories.

P. 13/lines 225-226: What do the authors mean on the statement that “the cluster approach created profiles that were more distinctive between each other”? Though this can be an interpretation of the larger Silhouette coefficient of the cluster analysis approach, it cannot be derived from Table 1 to which the authors refer here.

P. 13/lines 231-233 (similar problem with P. 16/lines 307-308): According to the authors “The most important finding in these first analyses is that although apparently similar with regard to homogeneity within groups, the two approaches differ in heterogeneity between the groups”. The results do not confirm this statement. If we compute an overall homogeneity of the two profiles by computing a weighted homogeneity of the median split and cluster categories we get 0.75 and 0.62 respectively, which reflects a similar difference between the two methods than the difference between the two Silhouette coefficients (0.59 and 0.68), used for assessing the distinctiveness of the profiles. This means that there is a nice consistency between the difference in homogeneity and the difference in distinctiveness of the two profiles.

P. 14/lines 257-260: Correct statement about the classification of the cases of the two profiles based on the results of module Exacon of ROPstat. However, a global measure (such as Rand index or the Adjusted Rand index) would be needed to describe the overall similarity of the two categorizations. These indices are available in the output of Exacon.

Additional comments

P. 5/line 36 (and also 9/ line 135): “analyses” should be replaced with “analysis”.

P. 6/lines 63-69: „Wilson and colleagues (1998) indicated that there is no significant correlation between positive affect and negative affect as measured by the Positive Affect and Negative Affect Schedule” The conclusion „In other words, these two dimensions that compose the affective system are independent from each other” is not correct, because in the case of a zero Pearson correlation there might still be a nonlinear dependency between the two variables.

P. 9/line 133: “and” should be dropped before “(cf. Lundh, 2015)”.

P. 9/line 137: “re-location” should be replaced with “relocation”.

P. 12/ line 210: “Ropstat” should be replaced with “ROPstat”.

P. 12-13/lines 219-220: “Silhouette Coefficient closer to 0 indicates that the groups overlap” cannot be correct in the present situation where the different subgroups of both the median split and the cluster analysis procedure are necessarily all nonoverlapping.

P. 13/lines 223-224: An expert reader of the paper would like to now how can the SD of a Silhouette coefficient be computed, e.g. for comparison purposes. A reference would be enough here.

P. 32-38/ Replace decimal commas with decimal points in Tables 1, 3, and 4.

·

Basic reporting

In this paper, the authors described their analyses on comparing two approaches to defining affective profiles: median splits and cluster analyses. The overall writing is clear, but there are some grammatical errors, and some sentences are long and redundant. Please fix the errors and shorten some sentences.

The targeted readership of this paper is supposedly researchers in psychology. To make the article more easily understandable to researchers in other fields, I suggest the following changes.

1. The Methods section in the abstract only contains the study design and the results. Please add some description about the analyses the authors used to reach the conclusions.

2. In line 66, please specify that "these dimensions" refer to positive affect and negative affect.

3. Please rephrase the sentence in lines 76-80. The current version is lengthy and vague.

4. In line 94, please change "along" to "along with".

5. In line 109, "during" is a bit vague. A better choice could be "found by".

6. What did the authors mean by "cut-off points found in the original study" in line 118? Do they refer to median splits found in the current small sample?

7. In line 121, is "by fiat" a commonly used phrase in psychology? How about "fiat" -> "arbitrariness"?

8. Incomplete sentence in line 133: "and (cf. Lundh, 2015)

9. In line 150, "this despite" is a grammatical error. "This" should be removed.

10. Please write down the formula for the calculation of Cronbach's alpha in line 196.

11. In Table 1, should the commas "," between two numbers be decimal points "."?

12. Please write down the formula for the homogeneity coefficient in line 216.

13. In lines 274-283, the authors listed the percentages of males and females in each affective profiles group. It would be better to list those percentages in Tables 3-4 than in the main text.

Experimental design

From my point of view, the difference between the affective profiles generated by median splits and clustering lies in the data distribution. If the data have a symmetric and unimodal distribution in a dimension, it is reasonable to use median splits in that dimension. If the data has a bimodal distribution that can be well separated into two clusters in a dimension, it is reasonable to use clustering in that dimension. Hence, I think the choice between median splits and clustering should be dimension-wise and data dependent. I would like to see some distribution plots of the study data in this paper. So we can have a better understanding of the comparison results.

Validity of the findings

About the K-means clustering, I wonder if the authors have tried multiple initial values to obtain the clustering results. A two-dimensional scatterplot will be necessary to see how well the K-means clustering performed.

In line 223, the authors used a z-test to test the difference of Silhouette Coefficient. Since z-test has an underlying Gaussian assumption, the authors need to justify why it is proper here.

In line 224, the authors wrote "equally homogeneity". "Equally" should be "equal". Besides, we don't see equal homogeneity coefficients between the two approaches. "Equal" can be replaced with "similar".

In the chi-square goodness of fit test in line 227, which approach did the authors treat as the expected values?

The sentence in lines 299-300 is incorrect. The authors wrote that "the cluster method ... assumes the existence of subpopulations, such as males and females". In fact, the clustering method uses no gender information, so it does not assume the existence of males and females. Instead, the clustering method assumes data clusters, which may or may not correspond to any real subpopulations such as males and females.

---

## Round 0.2 · Minor Revisions

· Academic Editor

Minor Revisions

The reviewers have some remaining concerns regarding the manuscript that must be addressed before a final decision can be reached.

·

Basic reporting

It is OK

Experimental design

It is OK

Validity of the findings

It is OK

Additional comments

 P. 6/lines 61: Omit comma from the end of "Schedule (Watson, Clark & Tellegen, 1988),."
 I miss the figure of scatter-plot of PA and NA
 P. 13/lines 229: “weighted average of the homogeneity coefficients” should be replaced with “weighted average of cluster homogeneity coefficients”
 Footnote 2 to page 13: What has “deff” to do with the concept of homogeneity coefficient (HC)? Here the authors should simply comment that HC of a cluster is the average of the pairwise differences of cases belonging to this cluster.
 Footnote 3 to page 13: This reference to the “Silhouette coefficient” has no sense. On the one hand it is incomplete (the Silhouette coefficient is the average of s(i) values for the whole sample), on the other hand without explaining the meaning of the a(i), b(i) components the reader will not understand anything. Suggestion: do not refer to the formula, but refer only to the source (Rousseeuw, 1987)
 P. 20/ lines 406: The authors write: “If the data have a symmetric and unimodal distribution in a dimension, it is reasonable to use median splits in that dimension” I would rather create here three categories with two cutoff points (e.g. with quartiles 1 and 3): one category in the middle and two on the tails.
 The reviewers may deserve to be mentioned in the “Acknowledgements”

·

Basic reporting

The authors' revised manuscript has been greatly improved from the previous version. I only have the following comments on this revised version.

1. In the Rebuttal letter, the authors wrote "In other words, these two dimensions that compose the affective system are linearly independent from each other." This statement is incorrect, as having a Pearson correlation as zero does not imply independence. Please change "linearly independent" to "uncorrelated".

2. Since K-means clustering can be trapped at local solutions (resulting in non-unique clustering results), trying multiple initial values is important for making sure that the 4 clusters the authors obtained in the paper is the best result. Does the ROPstat K-means function randomly pick initial values? If so, the authors should run that K-means function for multiple times and report the best clustering result.

3. The word "theorize" was used for many times in the paper. However, this is not a common word in modern scientific literature, and in many contexts in this paper, a more accurate word should be "assume".

4. In the abstract, the word "which" in "Although the question whether which approach is more appropriate than the other is still without answer", should be changed to "one".

5. In addition to median splits and clustering, are there other existing methods to create affective profiles?

Experimental design

The experimental design is reasonable.

Validity of the findings

Overall the results are valid. Please see my comments under "Basic Reporting".

---

## Round 0.3 · accepted · Accept

· Academic Editor

Accept

Thank you for addressing all feedback throughout the review process.